# Spatial Distribution of Aerosol Characteristics over the South Atlantic and Southern Ocean Using Multiyear (2004–2021) Measurements from Russian Antarctic Expeditions

Sergey M. Sakerin [1,*][iD], Liudmila P. Golobokova [2][iD], Dmitry M. Kabanov [1,*], Olga I. Khuriganowa [1,2], Viktor V. Pol'kin [1], Vladimir F. Radionov [3], Olga R. Sidorova [3] and Yuri S. Turchinovich [1]

[1] V.E. Zuev Institute of Atmospheric Optics, Siberian Branch, Russian Academy of Sciences, Academician Zuev Square 1, 634021 Tomsk, Russia; khuriganowa@mail.ru (O.I.K.); victor@iao.ru (V.V.P.); tus@iao.ru (Y.S.T.)

[2] Limnology Institute, Siberian Branch, Russian Academy of Sciences, Ulan-Batorskaya 3, 664033 Irkutsk, Russia; lg@lin.irk.ru

[3] Arctic and Antarctic Reserch Institute, 38 Bering Str., 199397 St. Petersburg, Russia; vradion@aari.ru (V.F.R.); olsid@aari.ru (O.R.S.)

[*] Correspondence: sms@iao.ru (S.M.S.); dkab@iao.ru (D.M.K.)

**Abstract:** Since 2004, we have carried out yearly measurements of physicochemical aerosol characteristics onboard research vessels at Southern Hemisphere high latitudes (34–72° S; 45° W–110° E). In this work, we statistically generalize the results from multiyear (2004–2021) measurements in this area of the aerosol optical depth (AOD) of the atmosphere, concentrations of aerosol and equivalent black carbon (EBC), as well as the ionic composition of aerosol. A common regularity was that the aerosol characteristics decreased with increasing latitude up to the Antarctic coast, where the aerosol content corresponded to the global background level. Between Africa and Antarctica, AOD decreased from 0.07 to 0.024, the particle volume decreased from 5.5 to 0.55 $\mu m^3/cm^3$, EBC decreased from 68.1 to 17.4 $ng/m^3$, and the summed ion concentration decreased from 24.5 to 2.5 $\mu g/m^3$. Against the background of the common tendency of the latitude decrease in aerosol characteristics, we discerned a secondary maximum (AOD and ion concentrations) or a plateau (aerosol and EBC concentrations). The obtained spatial distribution of aerosol characteristics qualitatively agreed with the model-based MERRA-2 reanalysis data, but showed quantitative differences: the model AOD values were overestimated (by 0.015, on average); while the EBC concentrations were underestimated (by 21.7 $ng/m^3$). An interesting feature was found in the aerosol spatial distribution in the region of Antarctic islands: at a distance of 300 km from the islands, the concentrations of EBC decreased on average by 29%, while the aerosol content increased by a factor of 2.5.

**Keywords:** aerosol; black carbon; ionic composition; spatial distribution; Southern Ocean; South Atlantic

## 1. Introduction

Atmospheric aerosol plays an important role in the processes of radiative transfers and mass exchange of different substances in the "continent–atmosphere–ocean" system. Because of the variety of aerosol physicochemical compositions, interaction processes, and strong variations, there is still uncertainty regarding aerosol's quantitative characteristics in a number of regions. Primarily, these are the high-latitude regions of the ocean, where regular measurements are barely possible (except at scarce island stations).

The spatial distribution of directly marine aerosol over the ocean is relatively uniform. Nonetheless, the results of real measurements indicate that aerosol characteristics are highly variable, even in remote oceanic regions (e.g., [1–4]). The geographic differences may be due to the specific features of the hydrometeorological conditions of the generation of marine aerosol; however, the main effect is due to the outflows of continental (dust, anthropogenic, and smoke) aerosol to the marine atmosphere. That is, the spatial nonuniformities of

aerosol over the ocean are determined primarily by the strength of the continental sources in adjoining regions and the predominant air mass circulations. As is known (e.g., [1–7]), the strongest effect is exerted by the trade wind and monsoon outflows of aerosols of different types in the Atlantic, Pacific, and Indian Ocean basins (e.g., [4–7]). The effect of continental aerosol is also manifested in the high-latitude regions of the ocean, but there are differences between the Northern and Southern polar zones. The atmosphere of the Arctic Ocean is subject to permanent aerosol outflows from Eurasia and North America (e.g., [8–11]). The main territory of the Southern Ocean is more than two to three thousand kilometers from the sources of continental aerosol. We also note that the aerosol content in the atmosphere of the Antarctic bordering the Southern Ocean is considered as the global background level. A change in this level serves as an indicator of global changes in the aerosol content of the Earth's atmosphere, such was the case after the strong volcanic eruptions of El Chichón and Pinatubo.

Aerosol studies in the polar zone of the Southern Hemisphere were initiated about half a century ago; however, they were carried out mainly at Antarctic stations [1,12–16]. The aerosol studies immediately over the Southern Ocean had long been episodic in character. That is, the results obtained reflected the aerosol properties and characteristics in specific situations, and not the average variability regularities. The implementation of the Fourth International Polar Year, associated with climate change, stimulated more active research into aerosol physicochemical characteristics in high-latitude regions, namely the disperse, ion, and elemental compositions of aerosol, the content of the absorbing substance (black carbon) in aerosol, the aerosol optical depth (AOD) of the atmosphere, etc. [2,5,17–25]. Despite the undoubted importance of the results obtained, they are still inhomogeneous with respect to the regions and studied characteristics. We note that, to obtain climatically significant data (average spatial distribution, and seasonal and interannual oscillations), of primary concern are not detailed studies, but rather multiyear homogeneous measurements, even if a limited set of aerosol characteristics is addressed.

The most abundant and homogeneous data obtained are those on the atmospheric AOD. The progress in AOD measurements was reached owing to the development of means of satellite (MODIS, MISR, etc.) measurements and to the use of quite simple portable sun photometers, i.e., Microtops II [26] and SPM, in marine expeditions [27]. The systematic accumulation of data from ship-based AOD measurements was also favored by the arrangement of the Maritime Aerosol Network component [28].

Since 2004, we have performed yearly measurements of the AOD, aerosol and black carbon concentrations, as well as the aerosol chemical composition on the route of research vessels from Europe to Antarctica. Based on multiyear data (until 2015), we analyzed the average latitude (with a step of 5°) variations in aerosol optical and microphysical characteristics in the East Atlantic from 60° N to 70° S [29]. In subsequent works, we suggested zonal empirical models of aerosols in the East Atlantic [30] and considered more comprehensively the average latitude behavior of aerosol characteristics over the ocean from the African to Antarctic coasts [31] and the specific features of the spatial distribution near Antarctic islands [32].

Taking into account the measurements from new expeditions (2018–2021), in this work we refine the specific features of the spatial distribution of atmospheric aerosol characteristics over the ocean in the latitude zone of 34–70° S. The refinement is motivated not only by the involvement of new data, but also by a change in the calculation techniques. In contrast to [31,32], in this case, we analyzed the volumes of particles in two (fine and coarse) aerosol fractions and the ionic composition. Moreover, we recalculated the results from measurements of the aerosol and black carbon concentrations using a homogeneous algorithm for the filtering and preliminary processing of the initial data [33]. Previously, the procedure of identifying and sorting out bursts (contaminated measurements) in the initial observation time series was carried out through visual data control; i.e., it was partly subjective in character.

## 2. Characterization of Expedition Measurements

Our aerosol studies in the southern part of the World Ocean were initiated in 2004 during the 19th cruise of the research vessel (RV) *Akademik Sergei Vavilov* [34]. Since 2005, the measurements have been conducted onboard the RVs *Akademik Fedorov* and *Akademik Tryoshnikov* in yearly Russian Antarctic Expeditions (RAE). Most of the RV cruises were carried out along the same route in the East Atlantic up to Cape Town, followed by transitions to the Antarctic stations Novolazarevsakya, Molodyozhnaya, Progress, and Mirny. The measurements in the latitude zone of 34–70° S were carried out in December–April, and in November–December in 2004. A fragment of the map with RV routes in the Indo–Atlantic sector of the ocean (45° W–110° E) is presented in Figure 1. Unique results were obtained in the International Antarctic Circumnavigation Expedition (ACE) program onboard the RV *Akademik Tryoshnikov* (62nd RAE) [35]. During this expedition, comprehensive studies of the nature of Antarctic islands, including the characteristics of atmospheric aerosol, were conducted for the first time (Figure 2).

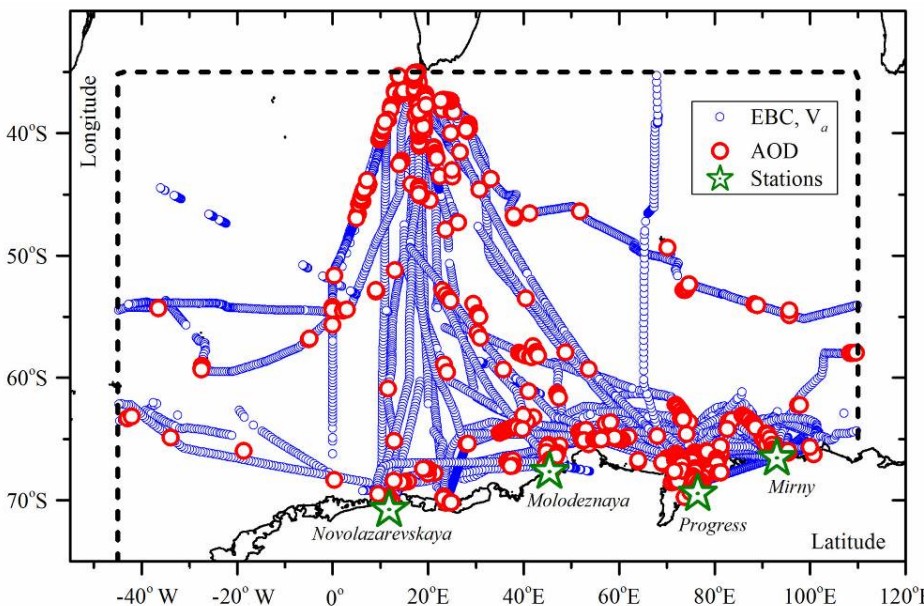

**Figure 1.** Locations/sites of the measurements of aerosol characteristics on the routes of the Antarctic expeditions in the Indo–Atlantic sector of the ocean (34–70° S; 45° W–110° E).

Table S1 (see Supplementary Materials) presents a list of the Antarctic expeditions and the numbers of days of measurements and collected samples of aerosol (*n*) in the Indo–Atlantic sector of the ocean. These expeditions conducted homogeneous measurements of the atmospheric AOD $\tau^a(\lambda)$, the number concentration of particles with radii of 0.15–5 µm in separate size ranges ($N_i$), and the mass concentration of light-absorbing substance (black carbon) in aerosol.

The atmospheric AOD was measured mainly by an SPM sun photometer (the wavelength range of 0.34–2.14 µm) [27] and partly by a Microtops II photometer [26]. The analysis was performed using the dataset obtained until 2016 [30], which was complemented with the AOD measurements in the last four expeditions (63rd–66th RAEs). Considering that fine and coarse aerosols differ in nature, the variations were analyzed for two AOD components: $\tau^a(\lambda) = \tau^f(\lambda) + \tau^c$, where $\tau^c$ is the coarse component caused by the quasi-neutral attenuation of light by large particles, and $\tau^f(\lambda)$ is the selective fine component of AOD. The $\tau^c$ component was determined from the AOD in the IR wavelength range (1.24–2.14 µm), and $\tau^f$ was calculated as a residual at the wavelength of 0.5 µm: $\tau^f_{0.5} = \tau^a$ (0.5 µm) $- \tau^c$ [29,36].

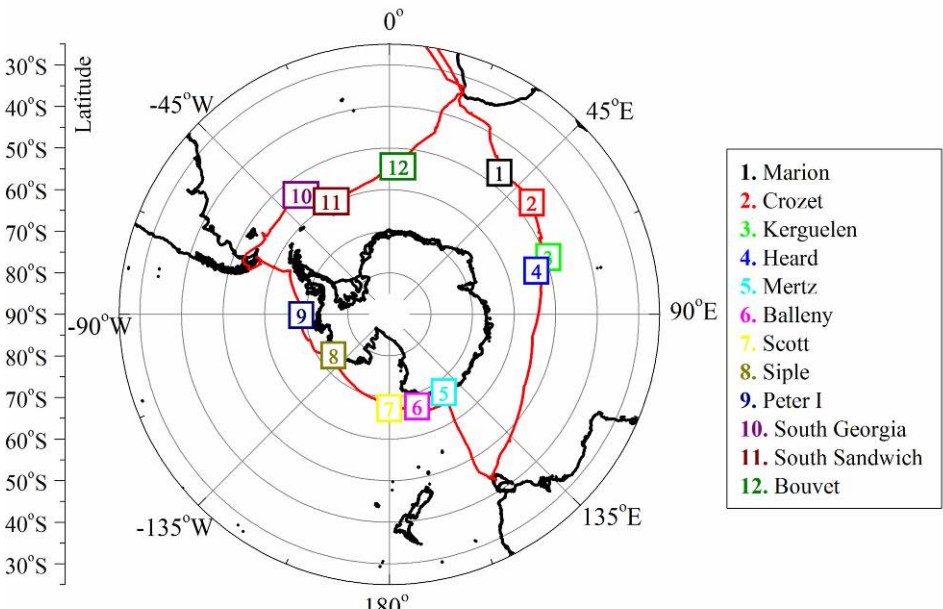

**Figure 2.** Geographic positions of islands on the route of the RV *Akademik Tryoshnikov* (62nd RAE).

The measurements of the concentrations of aerosol and light-absorbing substance were carried out using the photoelectric particle counters AZ-10 [37] or GRIMM 1.108 [38] and the aethalometer MDA [39]. The particle counters were calibrated using polystyrene latexes with known sizes. The aethalometer was calibrated using a generator of black carbon particles and the simultaneous concentration measurements were taken by the gravimetric method and by the aethalometer [40]. That is, the concentration of light-absorbing substance was measured in the equivalent of elemental black carbon. Taking into account the recommendations in [41], we will use the term EBC mass concentration.

The instruments were mounted onboard the ship at a height of about 15 m above the sea surface level. The concentrations of aerosol and black carbon were measured round–the–clock every hour. A single measurement cycle lasted for 10–20 min. The measured particle concentrations $N_i$ were used to calculate the volumes of particles in the fine ($V_f$) and coarse ($V_c$) aerosol fractions and the total volume was calculated as: $V_a = V_f + V_c$, where $V_{f(c)} = \Sigma\ 4/3\cdot\pi\cdot r_i^3\cdot N_i$ in the radius ranges of 0.15–0.5 μm and 0.5–5 μm, respectively. We note that the volume of particles is proportional to the aerosol mass concentration: $M\sim\rho\cdot V$, where $\rho$ is the density of the substance of the particles.

The measurements of the concentrations of $N_i$ and EBC contained short-term gaps and bursts (contaminated measurements) due to the effects of local sources of technogenic origin: smoke from the ship's funnel or polluted air from the ventilator shafts. A special algorithm [33] was used to filter the initial data to identify these defects as long as 3 h and recover the data. The issue of the data filtering was especially important under the conditions of low aerosol and black carbon concentrations, characteristic of the clean atmosphere of remote regions of the ocean.

A preliminary analysis of the purified data time series showed that it, nevertheless, still contained a small number (less than 2%) of bursts that cannot be explained by natural causes. These bursts in the aerosol and black carbon concentrations were mainly observed when the ship stopped near the Antarctic stations and were a few hours in duration. That is, they were also due to technogenic impacts, with a large probability. To eliminate these suspicious data, we applied an additional filtering procedure using the statistical criterion "three-sigma (*6*)".

The total amount of data, selected for the analysis of latitude variations in aerosol characteristics included: 2307 hourly average AOD values, 8652 EBC values, and 11197 $N_i$ values. Figure S1 (see Supplementary Materials) shows how the data were distributed over the months of measurement. The main part (75–82%) of the data was obtained in the period

from December to February (Southern Hemisphere summer period). Taking into account March and April, the data amount was 98–99%.

In addition to instrumental measurements of AOD, $N_i$, and EBC in the Antarctic expeditions, we also collected aerosol samples on filters for subsequent determination of the aerosol chemical composition. In this work, we considered only the ion composition of aerosol, for which the largest data amount was obtained. Using a pump-compressor, air was pumped through a Teflon filter PTFE (Advantec Toyo Roshi Kaisha, Tokyo, Japan) with a pore diameter of 0.8 µm and a size of 47 mm. To deposit enough material on the filters, the amount of pumped air was no less than 10 m$^3$ in volume. We used different pump types, so the pumping of air (collecting a single sample) varied from 12 to 24 h in duration.

Substances from the exposed filters were extracted using double-distilled water in an ultrasonic bath WUC-A02H (Daihan Scientific, Seoul, South Korea) for 30 min and filtered through a membrane filter with a pore diameter of 0.2 µm. In the filtrate thus obtained, we measured the value of pH using the pH meter Expert (Econix-Expert LLC, Moscow, Russia) and the concentrations of eight water-soluble ions: $Ca^{2+}$, $Mg^{2+}$, $Na^+$, $K^+$, $NH_4^+$, $NO_3^-$, $Cl^-$, and $SO_4^{2-}$. The ion composition of aerosol was determined on a reagent-free system ICS-3000 with an accuracy of up to 2–6%, approved by the U.S. Environmental Protection Agency (EPA). The generation of eluent allows the analysis of anions and cations at the level of their trace amounts (see [42,43] for more detail). The level of fluctuation noises, drift of zero signal, and the deviation of the output signal of the device were controlled using control solutions of $Na^+$ and $NO_3^-$ with a concentration of 10 mg/L [44].

A preliminary analysis showed that the ion concentrations obtained also exhibited episodic bursts caused by natural or technogenic factors (sea spray and polluted air from the ship). These bursts were observed most often when the vessel cruised slowly in ice and stopped near the Antarctic stations. Therefore, these data (about 3%) were eliminated using the statistical criterion "three-sigma (*6*)". The total number of samples selected for the statistical analysis is presented in Table S1 (see Supplementary Materials).

In addition to the absolute values of the concentrations of ions, we also considered the relative contents of different ions using the fractional factors *FM* [45] and the enrichment coefficients $K_i$ [46]. The fractional factors $FM_{cont}$ and $FM_{sea}$, related by the formula $FM_{cont} + FM_{sea} = 1$, make it possible to estimate the contribution of marine sources (primarily represented by $Na^+$ and $Cl^-$) to the ion composition of aerosol as compared to the contribution of other (conventionally "continental") sources. The fractional factor $FM_{cont}$ was calculated from the formula [45]:

$$FM_{cont} = [\Sigma (C^a_i - k_i^{sw} \cdot C^a_{Na})]/[\Sigma (C^a_i)] \tag{1}$$

where $C^a_i$ is the mass concentration of the *i*-th ion, $C^a_{Na}$ is the mass concentration of $Na^+$, $k_i^{sw} = (C_i^{sw}/C_{Na}^{sw})$ is the ratio of the concentration of the *i*-th ion to the concentration of $Na^+$ in seawater, and $\Sigma C^a_i$ is the summed concentration of all ions in the aerosol.

The second characteristic, i.e., the enrichment coefficient $K_i$, can be used to estimate the ion composition of aerosol with respect to seawater. The enrichment coefficients were calculated using the relationship between the concentrations of different ions, normalized by the concentration of $Na^+$, in the compositions of aerosol and seawater [46]:

$$K_i = (C^a_i/C^a_{Na})/(C^{sw}_i/C^{sw}_{Na}) \tag{2}$$

where $(C^a_i/C^a_{Na})$ is the concentration of the *i*-th ion with respect to $Na^+$ in the aerosol composition, and $(C^{sw}_i/C^{sw}_{Na})$ is the concentration of the *i*-th ion with respect to $Na^+$ in the composition of seawater. We note that a significant excess of the enrichment coefficient ($K_i \gg 1$) indicates that ions have a continental (including anthropogenic) origin.

## 3. Results and Discussion

Before analyzing the spatiotemporal variations, we reiterate the main sources of aerosol and black carbon, as well as the factors influencing their concentrations. The aerosol content in the atmosphere over the ocean is determined by two sources: (a) the generation of marine aerosol, which depends primarily on the wind velocity (e.g., [47,48]), and (b) the outflows of continental aerosol of different types. The effect of continental aerosol depends on the direction of predominant circulations, but, on the whole, it decreases with distance from land. In polar regions, one more factor has an effect [31]: changes in the relative area of ice and icebergs, which overlap (blocks) the source of marine aerosol. We note that the quantitative effect of the ice-covered ocean area on the generation of marine aerosol has not yet been well studied. Thus, the characters of the latitudinal variations in AOD, $V_f$, and $V_c$ were determined by the actions of three competing factors.

The main sources of the light-absorbing aerosol component are located in continental regions (except snow-covered Antarctica). Therefore, the concentrations of EBC should decrease to the global background level in the atmosphere of Antarctica with increasing latitude (receding from Africa).

### 3.1. Latitude Variations in AOD, Aerosol, and Black Carbon Concentrations

The latitude variations in the aerosol optical and microphysical characteristics ($\tau^a_{0.5}$, $\tau^f$, $\tau^c$, EBC, $V_f$, and $V_c$) were analyzed within the following coordinates: 34–70° S and 45° W–110° E. The average latitude behavior was calculated by averaging the aerosol characteristics with a step of 3° in latitude. The latitude variations in the AOD, concentrations of EBC, and volumes of aerosol particles $V_a = V_f + V_c$ are presented in Figures 3a, 4a and 5a.

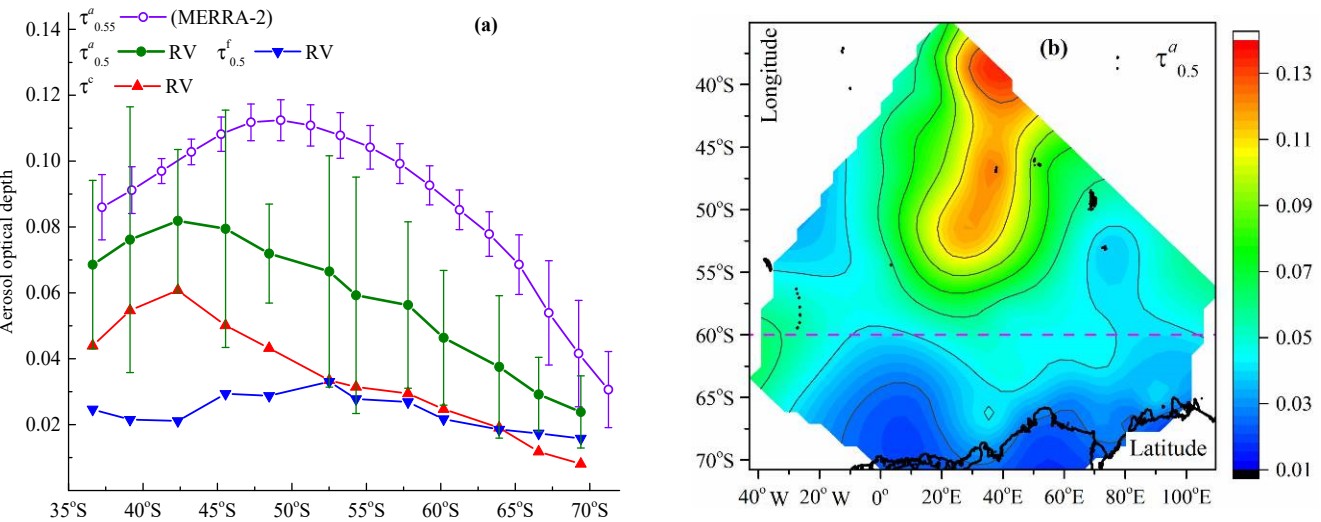

**Figure 3.** Average spatial distribution of the atmospheric AOD: (**a**) latitude behavior using ship-based (RV) measurements and MERRA-2 reanalysis data, and (**b**) latitude–longitude distribution based on the interpolation of ship-based data (dashed line indicates the Southern Ocean border).

In the latitude behavior of AOD (Figure 3a), there was a minor growth up to 42° S, followed by a linear $\tau^a_{0.5}$ decrease with a gradient of 0.022 per 10° of latitude. The characteristics of the latitude variations in AOD were due to the coarse component $\tau^c$, while $\tau^f$ varied in a narrow range of $0.02 \pm 0.005$. In the latitude behavior of the near-surface characteristics of aerosol (Figures 4a and 5a), three intervals with different variability characters were identified with a statistical significance: (1) a minor decrease in the average values of EBC, $V_f$, and $V_c$ to about 43° S, (2) a blurred maximum or a plateau in the latitude zone of 44–56° S, (3) followed by rapid EBC and $V_a$ drops up to Antarctica.

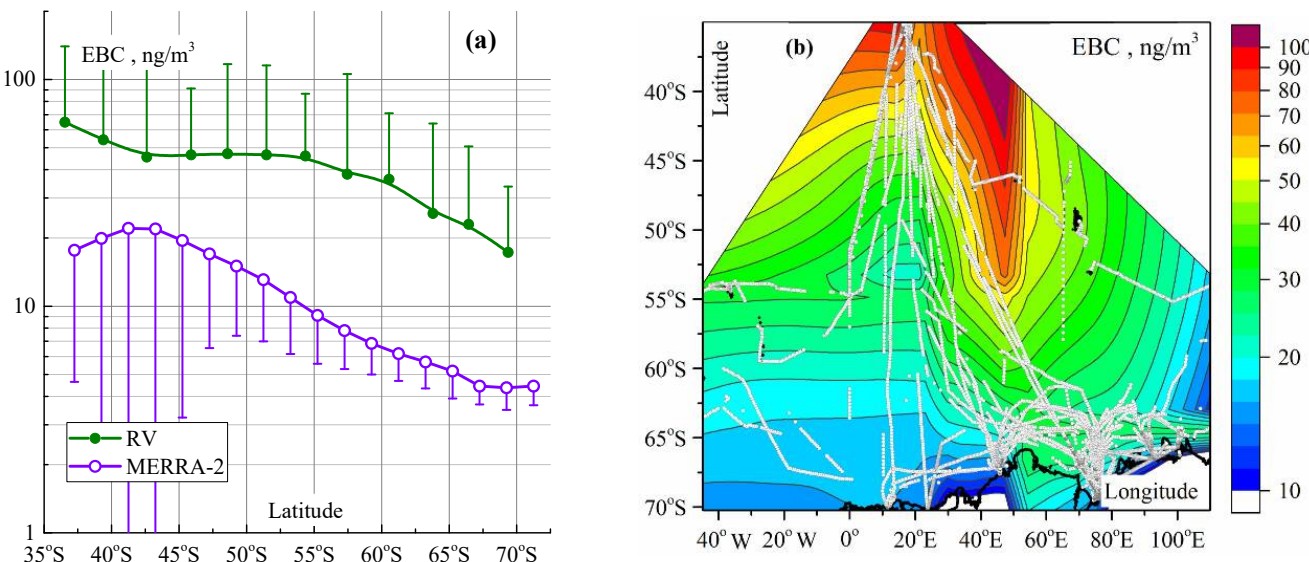

**Figure 4.** Average spatial distribution of EBC concentrations: (**a**) latitude behavior using ship-based (RV) measurements and MERRA-2 reanalysis data, and (**b**) latitude–longitude distribution based on the interpolation of ship-based data.

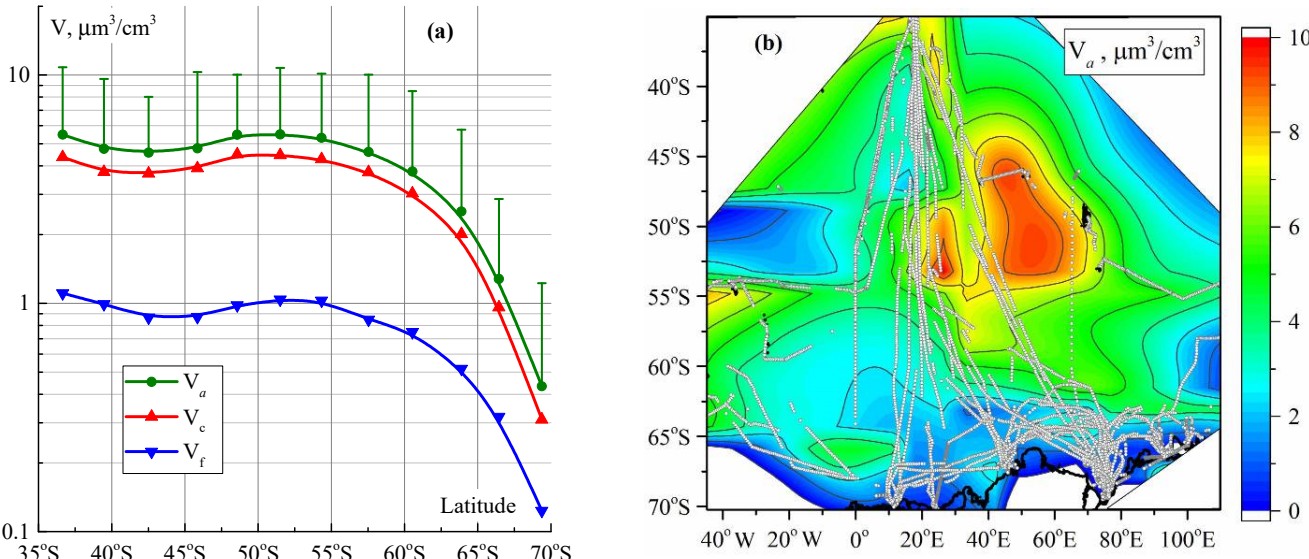

**Figure 5.** Average spatial distribution of volumes of aerosol particles: (**a**) latitude behavior of $V_a$, $V_f$, and $V_c$; and (**b**) latitude–longitude distribution of $V_a$ based on the interpolation of ship-based data.

The main decrease in the aerosol and black carbon concentrations begins at the latitudes of 55–60° S, where there is a border of drifting ice and the Southern Ocean (see the dashed line in Figure 4b). At these same latitudes, there is also the Subantarctic depression (low-pressure region), which separates the cyclonic transport belt at midlatitudes from the impact area of the Antarctic anticyclone. This conventional circulation boundary inhibits the further transport of mid-latitude air masses toward Antarctica.

We note that the aerosol content at latitudes φ > 45° S even started increasing, giving rise to a blurred maximum (Figure 5a). This maximum was due to the opposite actions of two factors: (a) the increasing generation of marine aerosol when passing to the zone of storm winds (φ > 45° S) and (b) a gradual (with the growing latitude) increase in the area of ice overlapping the source of marine aerosol. The light-absorbing component in the

composition of marine aerosol is insignificant. Therefore, the wind-driven increase in the content of marine particles does not lead to an increase in the concentrations of EBC.

The rapid decrease in the aerosol and black carbon contents at the latitudes of 55–70° S can be represented as a linear dependence: the gradient of the $V_a$ decrease is 0.34 $\mu m^3/cm^3$ per 1° of latitude, and the gradient of EBC decrease is 1.9 $ng/m^3$ per 1° of latitude. The decrease in the concentrations of EBC over the Southern Ocean can also be described by an exponential dependence, which was suggested for the Arctic atmosphere [9]. Independent of the extrapolation form, the decreasing gradient of the concentrations of EBC over the Southern Ocean is about a factor of two smaller than that in the Arctic.

Table 1 presents the average values and the standard deviations (±SD) of the aerosol characteristics in three latitude zones of the ocean: (a) in the immediate vicinity of Africa, (b) in the region of increased concentrations northward of the Southern Ocean, and (c) near the Antarctic coast. The total ranges of decreases in the aerosol characteristics from Africa to Antarctica were as follows: a factor of 2.9 for $\tau^a_{0.5}$, a factor of 3.9 for EBC, and a factor of 9.9 for $V_a$. Besides the latitude changes, contributions to the compositions of AOD and total particle volume were redistributed between coarse and fine aerosols. In the midlatitude ocean, the main contributor to AOD was coarse aerosol: $(\tau^c/\tau^a_{0.5}) > 60\%$ and $(\tau^c/\tau^f_{0.5}) = 2.6\text{–}3.3$. Near the Antarctic coast $(\varphi > 67°$ S), the contribution of coarse aerosol substantially decreased: $(\tau^c/\tau^a_{0.5}) = 35\%$ and $(\tau^c/\tau^f_{0.5}) = 0.86$. That is, fine aerosol started to play the key role, which is clearly seen from the intersection of the $\tau^f_{0.5}$ and $\tau^c$ curves in Figure 3a. The ratio of the volumes of particles in the two fractions changed in a similar way: the $(V_c/V_f)$ value was 3.95–4.64 at midlatitudes, while, at polar latitudes, the $(V_c/V_f)$ value decreased to 2.66.

**Table 1.** Average values and standard deviations (±SD) of the aerosol characteristics in three latitudinal zones of the ocean.

| Latitude Zone | $\tau^a_{0.5}$ | $(\tau^c/\tau^a_{0.5})$, % | $(\tau^c/\tau^f_{0.5})$ | $V_c$, $\mu m^3/cm^3$ | $V_f$, $\mu m^3/cm^3$ | $(V_c/V_f)$ | EBC, $ng/m^3$ |
|---|---|---|---|---|---|---|---|
| (a) 35–38° S | 0.069 ± 0.026 | 61 | 3.32 | 4.37 ± 4.49 | 1.11 ± 1.11 | 3.95 | 68.1 ± 75.7 |
| (b) 43–56° S | 0.076 ± 0.036 | 63 | 2.60 | 4.26 ± 4.34 | 0.92 ± 0.96 | 4.64 | 43.9 ± 55.5 |
| (c) >67° S | 0.024 ± 0.011 | 35 | 0.86 | 0.40 ± 0.91 | 0.15 ± 0.24 | 2.66 | 17.4 ± 17.1 |

This transformation of the aerosol disperse composition owes to the different lifetimes and transport distances of small and large particles. Fine aerosol is transported thousands of kilometers; moreover, it is generated in the atmosphere. Coarse aerosol is mainly local and depends on hydrometeorological conditions of generation. Correspondingly, in passing to the ice-covered zone of the ocean, large particles decrease faster in content than small ones.

The aerosol characteristics in Table 1, on the whole, are consistent with data from other authors (e.g., [2,5,16–20]). In this case, we kept ourselves from quantitative comparison, because the publications by other authors only presented average aerosol characteristics for separate expeditions in the Southern Ocean, which could differ from our multiyear data.

In addition to the average latitude behavior of aerosol characteristics, we also plotted the maps of their latitude–longitude distribution based on the Thin Plate Spline (TPS) (OriginPro 2016) interpolation [49] (this software is contained in the Origin package, http://www.OriginLab.com, accessed on 15 January 2022). Interpolation was performed on the basis of the average hourly measurements obtained for all years of measurements. The maps obtained (Figures 3b, 4b and 5b) give a clear and more comprehensive idea of the specific features of the spatial variations in the aerosol characteristics in the entire region. The plume of aerosol outflow from Africa, extending to about 55° S, clearly manifested in all characteristics. Of note is the maximum of aerosol content, localized at the latitude of 45–55° S (Figure 5b). The contributor to this maximum was the intense generation of marine aerosol in the zone of storm winds coupled with the outflow of continental aerosol from Africa. Over the Southern Ocean, the longitudinal inhomogeneities of aerosol

characteristics leveled out, and their latitudinal decrease towards the coast of Antarctica became the main regularity.

The maps of the latitude–longitude distribution of aerosol characteristics qualitatively agreed with the average latitude behaviors (Figures 3a, 4a and 5a). However, the interpolated values of the aerosol characteristics could not be used for quantitative estimates because the measurement sites were distributed with different densities at diverse latitudes. As can be seen from Figure 1, the measurement sites were located within a sector expanding from the southern tip of Africa (~18° E) toward Antarctica (45° W–110° E).

### 3.2. Spatial AOD and EBC Distributions Using MERRA-2 Reanalysis Data

Comparison with any independent data was important to confirm the regularities of the aerosol spatial distribution. The scientific literature provides no statistical generalizations based on the multiyear yearly measurements of aerosol characteristics in marine expeditions. Therefore, we carried out a comparison with the model-based MERRA-2 reanalysis data [50–52], which are the product of assimilating the results from the atmospheric AOD (AERONET or MODIS) measurements, models of meteorological fields, 3D distributions of aerosol of different types, and air mass transports. Among the reanalysis products, there are AOD and EBC with a spatial resolution of $0.5° \times 0.625°$, which are freely available [53]. The MERRA-2 reanalysis data are especially important for high-latitude regions [54,55], in which the actual measurements are still insufficient.

The model spatial distributions of AOD and EBC were calculated using the monthly average values of these characteristics for the period of ship-based measurements: December–April 2004–2021. For aerosol climatology, it is important that MERRA-2 reanalysis products are regular, multiyear, and spatially homogeneous. Figure 6 presents the maps of the average spatial distributions of AOD and EBC in the study region (34–70° S; 45° W–110° E) calculated using MERRA-2 data. A comparison with analogous maps, presented in Figures 3b and 4b, showed a qualitative similarity: AOD and EBC decreased from Africa toward Antarctica, with the trace of aerosol outflow from the continent being clearly manifested. Some differences may have been due to the different amounts and distribution densities of the model (MERRA-2) and experimental (RV) data. At the same time, the main difference was that the maps of the model AOD distributions showed a band of increased atmospheric turbidities at the latitudes of 43–53° S (Figure 6a), in contrast to the localized plume in Figure 3b. This difference can be clearly seen in the plots of the average latitude AOD dependences calculated using the MERRA-2 reanalysis data and the actual measurements (Figure 3a). The model-based AOD values were larger than the experimental ones at all latitudes. The largest deviation from the experimental AOD was 0.045, or 75%. There are grounds to believe that this difference was due to the overestimated satellite (MODIS) AOD values, which were used in the MERRA-2 reanalysis. A few publications have already discussed the issue of the anomalously high AOD retrieved from satellite observations in the latitude zone of 40–60° S [56–58]. Different hypotheses regarding the imperfection of the algorithms of cloud screening and the models used were suggested to explain this artifact. In our opinion [59], the main cause was that the AOD retrieval algorithm neglected the effect of the increased albedo of the sea surface partially covered by icebergs and ice.

The latitude dependences of the black carbon concentrations showed an opposite relationship between the model and experimental data (Figure 4a): the experimental EBC ($\varphi$) curve was 8–36 ng/m$^3$ higher than the modeled one. This difference is unexplained. There could have been a contribution from the ship's technogenic sources that survived in the data from expedition measurements (despite the burst filtering applied) or the model calculation's underestimated EBC values. The quantitative differences between the model and experimental AOD and EBC values could have arisen due to the different densities of the spatial distributions of the two types of data. Therefore, we additionally compared the data from reanalysis and ship-based measurements matched to be coincident in time ($\pm 1$ h) and collocated in coordinates ($\pm 1$ degree). The numbers of the joint data were 2307 AOD

values and 8652 EBC values. The comparison of the actual measurements and model calculations of AOD showed quite a high correlation (R = 0.65). The average difference between the data $\Delta\tau = [\tau^a_{0.5}$ (MERRA-2) $- \tau^a_{0.5}$ (RV)] was $0.015 \pm 0.019$. More than 90% of the $\Delta\tau$ values were within the range from $-0.015$ to $0.045$ (Figure 7a). The interrelation between the measured and modeled EBC values was weak, but statistically significant. In 90% of the cases, the differences $\Delta_{EBC} = [$EBC (MERRA-2) $-$ EBC (RV)$]$ were within the range from $-65$ to $20$ ng/m$^3$, the average being $\Delta_{EBC} = -21.7$ ng/m$^3$ (Figure 7b). That is, the MERRA-2 reanalysis data overestimated AOD and underestimated EBC, but the average difference was comparable to the error of measuring and modeling these characteristics. A comparative analysis of the reanalysis data and ship-based measurements of aerosol characteristics was presented in more detail in [60].

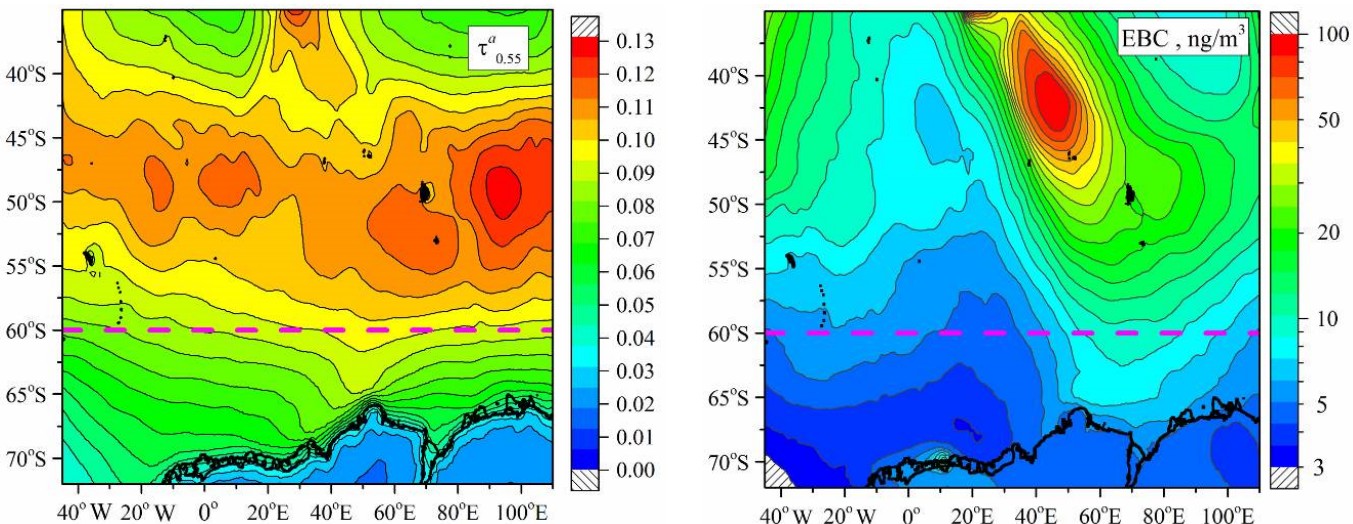

**Figure 6.** Average spatial distribution of AOD and black carbon concentrations according to MERRA-2 reanalysis data.

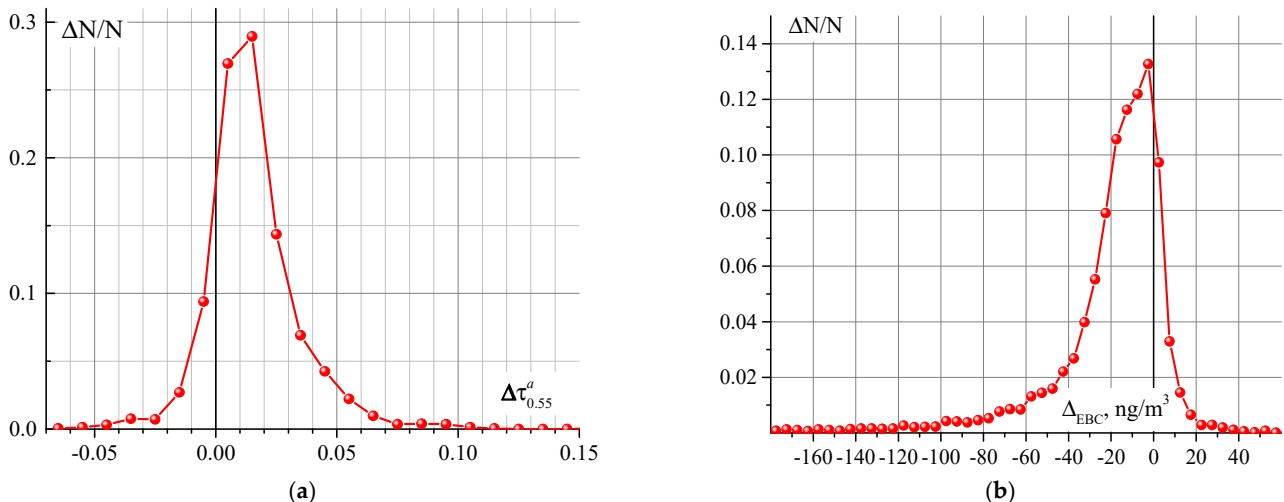

**Figure 7.** Frequency histogram of the differences between the model (MERRA-2) and experimental (RV) values of AOD and EBC ($\Delta\tau$ and $\Delta_{EBC}$). (**a**) $\Delta\tau^a_{0.55}$ and (**b**) $\Delta_{EBC}$, ng/m$^3$.

### 3.3. Estimates of Interannual Variations in AOD and EBC over the Southern Ocean

We already indicated above the large spatiotemporal variations in aerosol characteristics at midlatitudes caused by the outflows of continental aerosol (see Figures 3–6). Therefore, the multiyear variations were analyzed for the more homogeneous atmosphere

of the Southern Ocean (φ > 60° S). We note that precisely in the Southern Ocean in the summer period, we obtained the largest amount of data: more than 90% of AOD, 75% of $V_a$, and 72% of the concentrations of EBC. The interannual variations were estimated from three-month (December–February) average values of AOD, $V_a$, and EBC. Considering that the MERRA-2 reanalysis data and ship-based measurements systematically differed (see Figure 7), the calculations were carried out using normalized* values of each aerosol characteristic (marked with an asterisk), such as $\tau^* = (\tau_i / \overline{\tau})$, where $\tau_i$ is the three-month average AOD value in the *i*-th year and $\overline{\tau}$ is the multiyear average.

Figure 8a–c present the interannual variations in the normalized aerosol characteristics over the Southern Ocean obtained using data from ship-based (RV) measurements and MERRA-2 reanalysis. Statistically significant (at the 0.05 level) AOD growth by an amount of 0.016 over the period analyzed manifested in the reanalysis data. The concentrations of EBC also showed an increasing tendency, but they were below the statistical significance level. There was no trend component in the ship-based AOD and EBC measurements. The trend in the ship-based data was difficult to identify due to large interannual oscillations: the coefficients of variations of AOD, EBC, $V_f$, and $V_c$ were 27%, 50%, 74%, and 72%, respectively. The reanalysis data show about two times smaller relative variations: the variation coefficients of AOD and EBC were 12% and 21% respectively. The increased variations in the ship-based values of aerosol characteristics were due to the substantially smaller spatial averaging scale and fewer data, as well as gaps in the measurements during separate years.

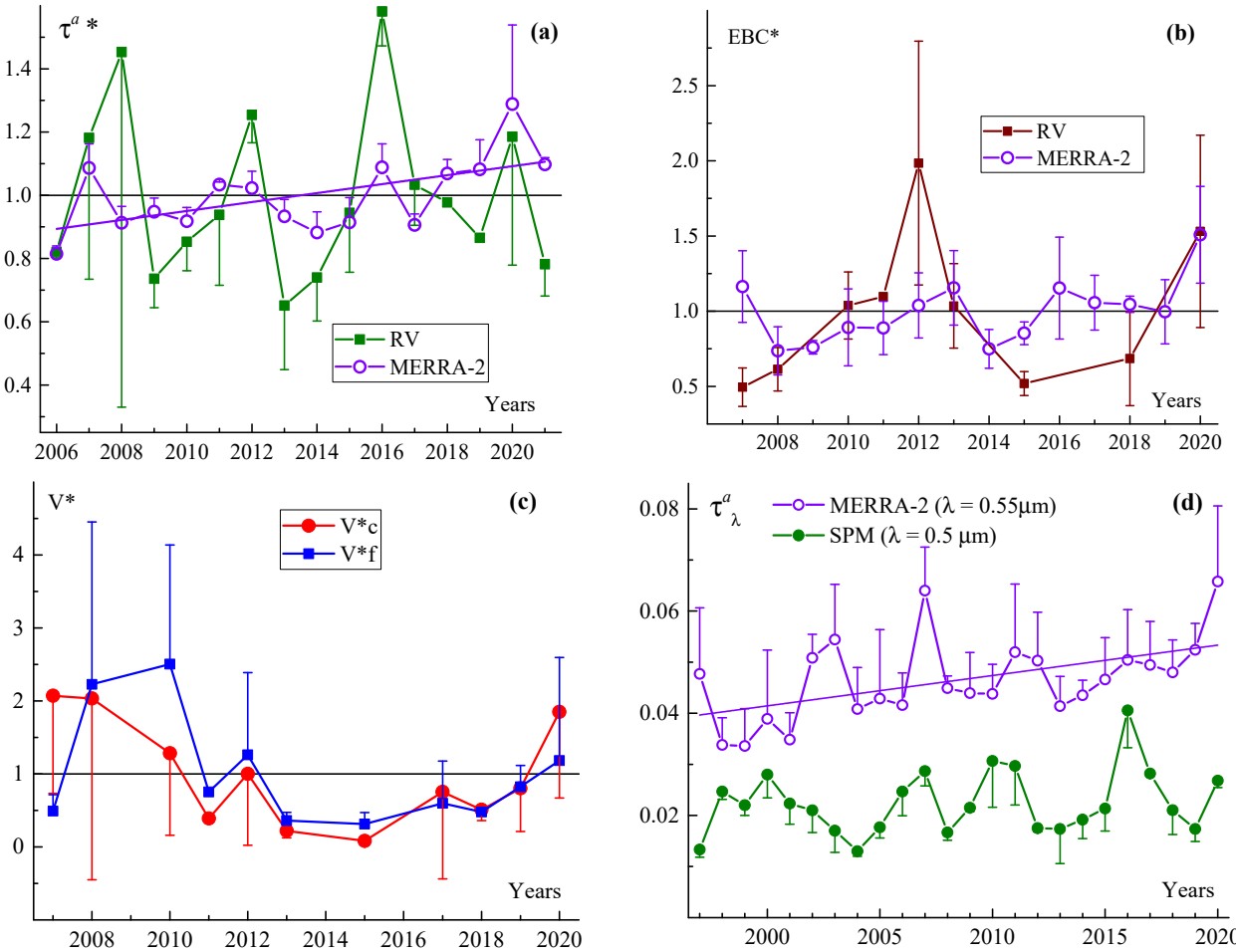

**Figure 8.** Interannual variations in aerosol characteristics over the Southern Ocean, using the data from MERRA-2 reanalysis and ship-based (RV) measurements: variations in (**a**) $\tau^a$ *; (**b**) concentrations of EBC*; (**c**) particle volumes $V_f^*$, $V_c^*$; (**d**) and the absolute AOD values in the region of the Mirny observatory, using data from SPM photometer [61] and MERRA-2 reanalysis.

There are some doubts about the positive AOD trend over the Southern Ocean from the MERRA-2 reanalysis data (slanted line in Figure 8a), which was not manifested in the ship-based data. This contradiction may either be due to the modeling (MERRA-2) inaccuracy or to the small period and regularity of ship-based measurements. To resolve this issue, we considered a longer-term (1997–2020) time series of AOD observations at the Mirny observatory, located on the Antarctic coast (66.6° S; 93° E) [61]. The data of the model calculations in this region showed a statistically significant AOD increase from 0.040 to 0.053 in 23 years (slanted line in Figure 8d), while the data from the photometric measurements indicated that the average value of AOD (0.5 µm) remained within 0.023 ± 0.006 (no trend). The systematic AOD overestimation (see Figure 7a) and the presence of a positive trend in the MERRA-2 data, inconsistent with the actual data, indicated that a small correction of the model calculations in this region was required.

However, a more interesting feature is that a periodic component was manifested in the multiyear AOD variations in the region of Mirny (Figure 8d), as well as over the Southern Ocean (Figure 8a,d). The periods of AOD oscillations, both in the reanalysis and photometric data, were, on average, 4.8 years. We note that, after 2005, the AOD oscillations using data from reanalysis and actual measurements became coordinated in character. In work [61], it was shown that these AOD oscillations were consistent with the change in the Antarctic Oscillation Index that characterized the periodic shifts of the wind belt and influx of marine aerosol toward Antarctica.

### 3.4. Spatial Variations in Aerosol and Black Carbon Concentrations near Antarctic Islands

From the maps presented in Figures 3b, 4b, 5b and 6, it can be clearly seen that the spatial distribution of aerosol at high latitudes became more uniform. The leveling off of the meridional distribution of the aerosol characteristics over the Southern Ocean was due to its remoteness from the continents and constantly acting westerly–easterly air transport. As a consequence of this, the average aerosol characteristics in the Atlantic, Indian, and Pacific sectors of the Southern Ocean (>55° S) barely differed [32]. However, on the scale of hundreds of kilometers, specific features may manifest in the spatial distribution of aerosol in the region of Antarctic islands. The prerequisites for this are the differences in the meteorological conditions, relief, and type of the underlying surface on which aerosols of different (marine, soil, and organic) types are generated. The participation in the International Antarctic circumnavigation expedition [35] with the visitation of 12 islands made it possible to estimate the effect of the "island" factor on the spatial variations in the aerosol and black carbon concentrations.

To estimate the effect of continental aerosol in the atmosphere of the Southern Ocean, we considered variations in the concentrations of $V_a$ and EBC with decreasing/increasing separation distance from the following Antarctic islands (see Figure 2): (1) Marion; (2) Crozet; (3) Kerguelen; (4) Heard; (5) Mertz; (6) Balleny; (7) Scott; (8) Simple; (9) Peter I; (10) South Georgia; (11) South Sandwich; and (12) Bouvet. Part of these islands is covered by moss and lichen, and the other part is covered by ice and snow. On some islands, there are polar stations with small personnel; the others are uninhabited, but are visited sometimes by scientific expeditions, tourists, and fishers.

The total amount of data obtained at a distance of up to 300 km from the Antarctic islands was 727 hourly values of EBC and 772 values of $V_a$. Figure 9 shows how EBC and $V_a$ varied as functions of the distance to the islands. Using different colors, the legend on the right indicates the islands near which the measurements were carried out, as well as the increasing (+) or decreasing (−) tendencies of the aerosol characteristics. The boxed formulas in the figures are equations of linear regression, giving the average (for all islands) dependence of concentrations on the distance.

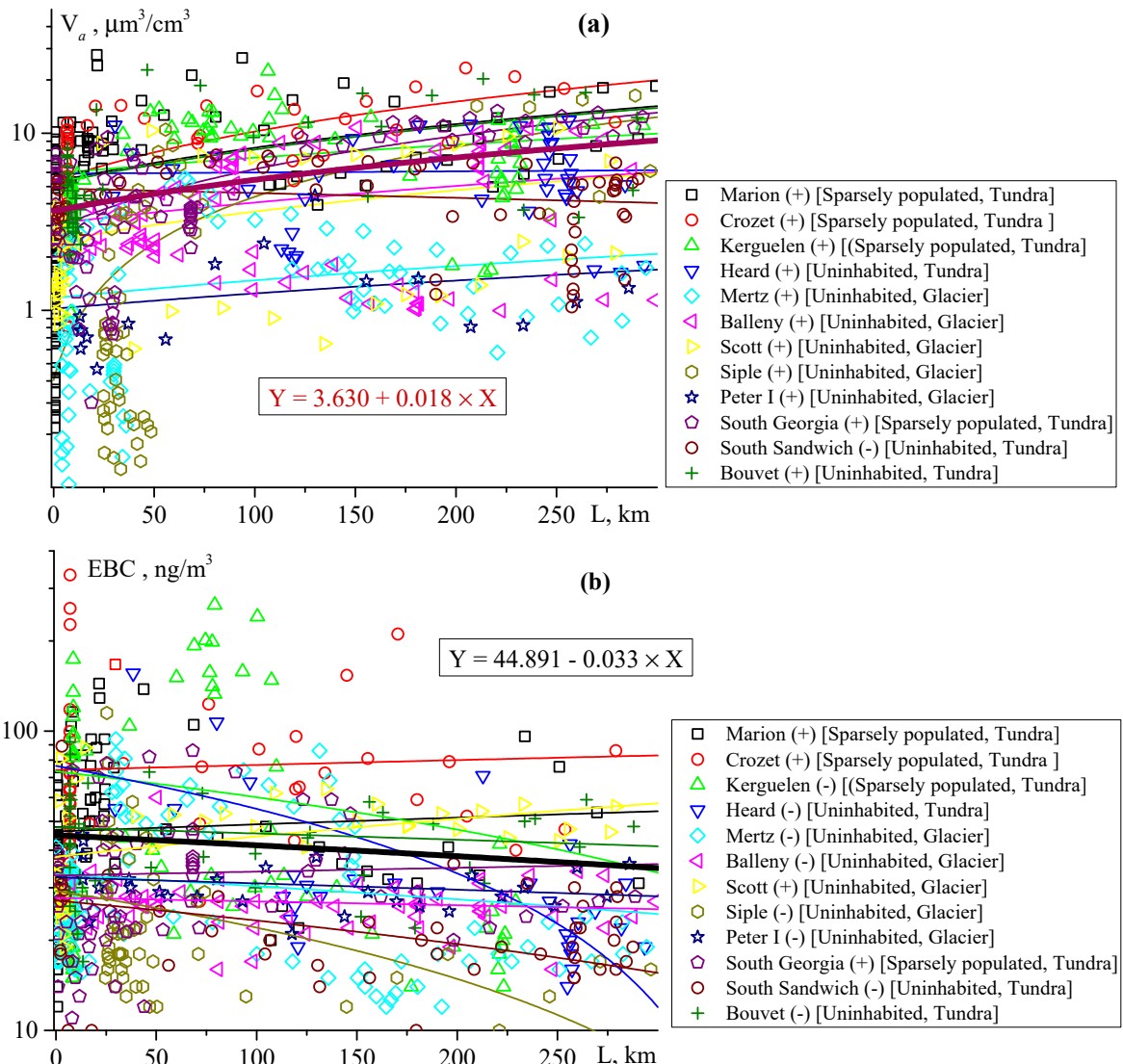

**Figure 9.** Changes in concentrations of (**a**) black carbon and (**b**) particle volumes with increasing separation distance from Antarctic islands.

The calculations showed that the contents of aerosol and black carbon changed in opposite ways as the distance from the islands increased to 300 km: the volumes of particles in both fractions ($V_f$, $V_c$) increased by about a factor of 2.5, while the concentrations of EBC decreased by 29% (or at 14 ng/m$^3$). This character of variation was observed in the region of eight (out of twelve) islands for the concentrations of EBC and in the region of eleven islands for particle volumes. The growth of the particle volumes was associated evidently with the more intense generation of marine aerosol in the open ocean as compared to the coastal zone. (Relief, and especially mountains, slow down wind in the region of the islands). The statistically significant decrease in the black carbon concentrations with the distance from the islands is more difficult to explain.

The main sources of light-absorbing aerosol (black carbon) are the products of fuel and biomass combustion (e.g., [62]). However, no smoke from vegetation burning was observed in the region of the islands. The conjecture on the possible anthropogenic impact (heating systems, etc.) has not been confirmed either: the concentrations of EBC decreased mainly with the distance from uninhabited islands. Moreover, in the region of three inhabited islands (Crozet, Marion, and South Georgia), the concentrations of EBC even increased with receding, contrary to the action of the anthropogenic factor. Seemingly, in this case, there

was an action of another absorbing aerosol component generally called "brown carbon", the quantitative contribution of which is still less studied. The sources of brown carbon are not only resinous combustion materials, but also diverse organics, i.e., humic substances, isoprenes, bioaerosols, etc. [63–65].

A common feature of the Antarctic islands is the large rookeries of penguins, pinnipeds, and sea birds. It is logical to speculate that the organic products of their life activities, as well as rotting algae on the coast, are the sources of emissions of light-absorbing substances into the atmosphere. As a consequence, the concentration of EBC measured in the region of the islands turned out to be higher than that in the remote regions of the ocean.

*3.5. Latitude Variations in Ion Composition of Aerosol*

The average latitude behavior of the concentrations of different ions ($C_i$, µg/m$^3$) and their sum ($C_\Sigma$) was calculated through data averaging within three- and five-degree latitude zones ($\Delta\varphi = 3°$ and $5°$) using two approaches. In the first, simplest version (a), the measured ion concentrations were referenced to the average coordinates (latitude) of each aerosol sample. In the second variant (b), we carried out a statistical leveling-off of the data with different sampling durations within the unit latitude zones $\Delta\varphi$. We first compiled a set of the hourly average * values of the concentrations over each sampling hour referencing the corresponding latitude. (In this case, the hourly average * concentration was assumed to differ weakly from the concentration obtained over the entire sampling period). Then, the hourly average concentrations were used to calculate the average latitude behavior of $C_i$ ($\varphi$) with spatial averaging over $3°$ or $5°$.

A comparison of the different variants of the calculations showed that the latitude variations in the ion concentrations were identical in character. As an example, Figure 10 presents the latitudinal behaviors of the summed concentrations of all ions $C_\Sigma$ ($\varphi$) calculated using two variants (a and b) of calculation and with the different latitude resolutions $\Delta\varphi$. Considering that the latitude variations in ion concentrations differ little between the diverse variants of the calculations, in the following, we will consider only the results from the second variant (b) with the latitude resolution $\Delta\varphi = 3°$. The summed ion concentration in this variant first increased a little (up to $42°$ S) and then decreased from 24.5 to 2.5 µg/m$^3$ almost linearly.

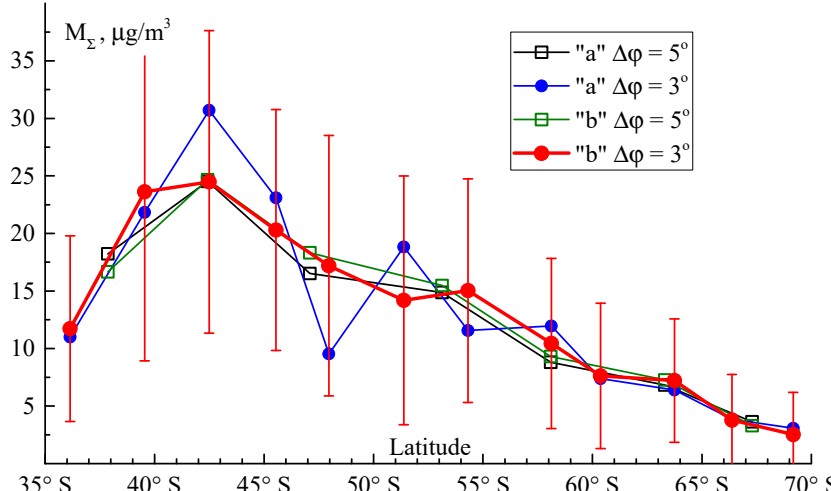

**Figure 10.** Latitude change of the summed concentration of ions in aerosol composition over the ocean between Africa and Antarctica in different variants of calculation (a and b) and ($3°$ and $5°$) averaging.

Figure 11 presents the average latitude behaviors of the concentrations of individual ions $C_i$ ($\varphi$) in the aerosol composition over the ocean between Africa and Antarctica. In order not to overburden the plots of the latitude behaviors of $C_i$ ($\varphi$), the error bars,

indicating the standard deviations (SD), are shown for just one variant of the calculation (b; $\Delta\varphi = 3°$) in Figure 10a and only for the upper and lower dependences $C_i(\varphi)$ in Figure 12.

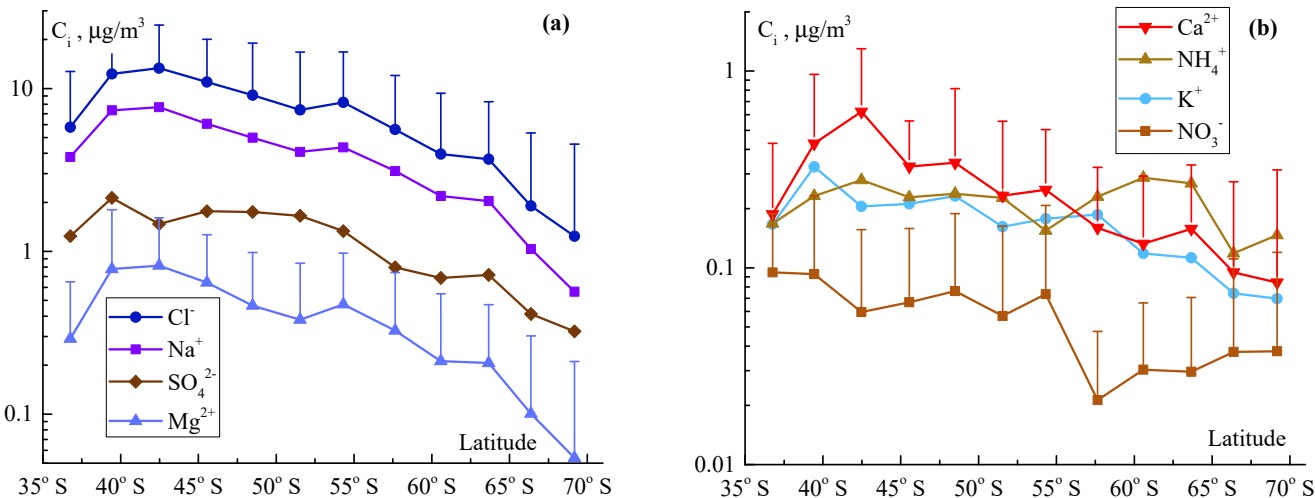

**Figure 11.** Latitudinal change in the concentrations of different ions in aerosol over the ocean between Africa and Antarctica. (**a**) $Cl^-$, $Na^+$, $SO_4^{2-}$, $Mg^{2+}$ and (**b**) $Ca^{2+}$, $NH_4^+$, $K^+$, $NO_3^-$.

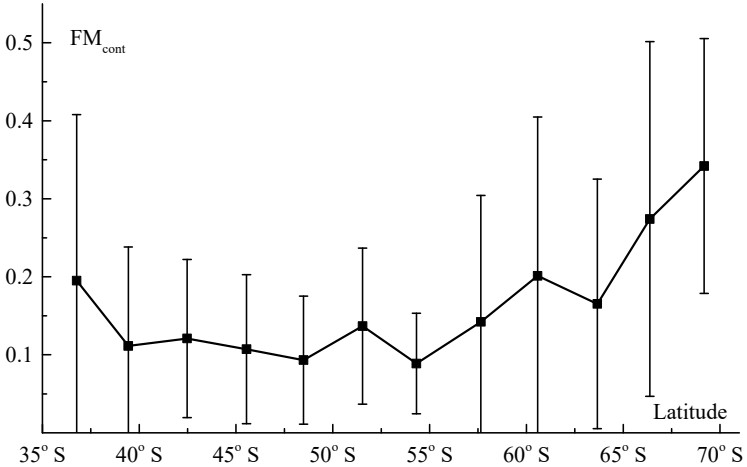

**Figure 12.** Latitudinal change of the fractional factor.

On the whole, the concentrations of all ions decreased from Africa to Antarctica (Figures 10 and 11). An analogous tendency was also observed in the variations of the concentrations of the other components of the aerosol chemical composition [21–23] and the characteristics considered in Section 3.1. In addition to the common tendency, the latitude variations of ions $Na^+$, $Cl^-$, $Mg^{2+}$, $K^+$, and $Ca^{2+}$ exhibited, though minor, statistically significant growth of the concentrations with growing distance from Africa (at latitudes up to ~40° S). It may appear strange that this growth of the ion concentrations was coupled with the decrease in the aerosol content (see Figure 4a). However, these are nevertheless different characteristics: a decrease in the particle volumes $V_f$ and $V_c$ may be accompanied by the enrichment of aerosol by some ions and, in particular, marine ones.

We turn our attention to the characteristics of the latitude behaviors of the $NH_4^+$ and $NO_3^-$ concentrations that were inconsistent with the common regularity of other ions. The peculiar features of the latitude variations in the $NH_4^+$ and $NO_3^-$ ions possibly stemmed from the biological productivity of oceanic waters and, as a consequence, from the nitrogen cycle between the ocean and the atmosphere [66,67]. At the same time, there are doubts about the significance of the features manifested, because the latitude variability ranges

of ions $NH_4^+$ and $NO_3^-$ were narrower than the standard deviations in separate latitude zones.

Despite the latitude variations (Figure 11), the distribution of ions with respect to the concentrations remains unchanged. The highest concentrations were exhibited by the ion $Cl^-$, and the ions $Na^+$ and $SO_4^{2-}$ were ranked next. The ions $Mg^{2+}$, $Ca^{2+}$, $K^+$, $NH_4^+$, and $NO_3^-$ were in the group with low concentrations. Table 2 presents the average concentrations of ions in the aerosol composition in three latitude zones of the ocean: near Africa (35–38° S), in the region of the maximum of the concentrations (38–44° S), and near Antarctica (>68° S). From these data it follows that the largest latitude variability range was observed in ions of marine origin: $Cl^-$, $Na^+$, and $Mg^{2+}$. Their concentrations decreased by a factor of 10–15. The concentrations of the ions $SO_4^{2-}$, $Ca^{2+}$, and $K^+$ changed by a factor of 4–6.

**Table 2.** Average values and standard deviations ($\pm$SD) of water-soluble ion concentrations ($\mu g/m^3$).

| Latitude Zone | $Cl^-$ | $Na^+$ | $SO_4^{2-}$ | $Mg^{2+}$ | $Ca^{2+}$ | $K^+$ | $NH_4^+$ | $NO_3^-$ |
|---|---|---|---|---|---|---|---|---|
| 35–38° S | 5.79 ± 6.94 | 3.79 ± 3.92 | 1.24 ± 1.16 | 0.29 ± 0.36 | 0.19 ± 0.24 | 0.17 ± 0.21 | 0.17 ± 0.23 | 0.10 ± 0.09 |
| 38–44° S | 12.76 ± 11.95 | 7.50 ± 6.95 | 1.86 ± 2.01 | 0.80 ± 0.93 | 0.52 ± 0.61 | 0.28 ± 0.37 | 0.25 ± 0.27 | 0.08 ± 0.13 |
| >68° S | 1.24 ± 3.32 | 0.57 ± 1.46 | 0.32 ± 0.48 | 0.054 ± 0.16 | 0.084 ± 0.23 | 0.07 ± 0.09 | 0.15 ± 0.22 | 0.04 ± 0.08 |

Figure 12 shows the latitude variations in the fractional factor $FM_{cont}$. In remote regions of the ocean, the fractional factor $FM_{cont}$ varied in the range of low values 0.12 ± 0.04. That is, marine sources mainly contributed to the formation of the ion composition of aerosol ($FM_{sea} \approx 0.88$). The increase in $FM_{cont}$ to 0.2 when approaching Africa does not need comments. A less obvious fact is that $FM_{cont}$ increased to a far larger value (0.34) near Antarctica.

The $FM_{cont}$ increase itself at latitudes $\varphi > 58°$ S may be because the underlying surface of the coast of Antarctica in the summer period becomes partially free of snow (ice), thus serving as the source of continental aerosol. The smaller $FM_{cont}$ values near Africa, as compared to Antarctica, could be due to the methodic factor of sampling. In the first case (near Africa), the measurements started a few tens of kilometers from the continent, with the distance from the continent rapidly growing to hundreds of kilometers. That is, the main sampling period corresponded to the large (200–400 km) distance from Africa, at which the effect of continental sources was significantly weakened. In the second case, samples were collected directly in the coastal zone (near the coasts) of Antarctica. That is, as compared to Africa, the source of continental aerosol, though being weaker, was closer.

Figure 13 presents the latitude variations in the enrichment coefficients for five key ions: $Cl^-$, $Mg^{2+}$, $K^+$, $SO_4^{2-}$, and $Ca^{2+}$. The enrichment coefficients for the ions $Cl^-$ and $Mg^{2+}$ at all latitudes were close to unity, i.e., the content of these ions was about the same as that in seawater. Two intervals with different latitude dependences manifested on the $K_i$ ($\varphi$) plots for the ions $SO_4^{2-}$, $Ca^{2+}$, and $K^+$. The enrichment coefficients in the first interval (35°–57° S) varied in the region of quite low values $K_i$ = 0.8–2.2 (i.e., the ion composition was close to that of seawater). However, at higher latitudes ($\varphi > 57°$ S), the enrichment coefficients for $SO_4^{2-}$, $Ca^{2+}$, and $K^+$ increased to 3.6–5.6. The growing enrichment coefficient when approaching Antarctica can be explained by the increasing influence of continental sources located on the Antarctic coast, with a simultaneously decreasing contribution from marine sources. The latter is favored by the growing area of ice-covered sea surface and by the decreasing salinity of the ocean.

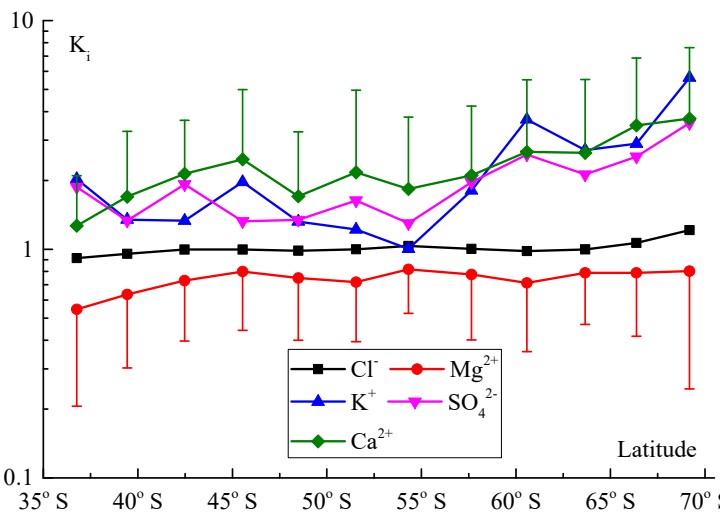

**Figure 13.** Latitudinal change of the enrichment coefficients of different ions.

### 4. Summary and Conclusions

Data from multiyear (2004–2021) expedition studies were used to analyze the average spatial distribution and interannual variations of the aerosol characteristics in the southern part of the World Ocean (34–70° S; 45° W–110° E). The regularities of the spatiotemporal variations in AOD and EBC over the ocean were compared with the data from the model (MERRA-2 reanalysis) calculations. The results of the studies made it possible to draw the following main conclusions.

A common regularity of the spatial distribution of aerosol characteristics was that the average AOD, $V_a$, and EBC values decreased from Africa up to Antarctica. The latitude variations of AOD were determined by the coarse component and were characterized by a minor increase up to 42° S, followed by a decrease to $\tau^a_{0.5}$ = 0.024 near Antarctica. The latitude behavior of the near-surface aerosol characteristics contained three characteristic intervals: (1) a minor decrease in the average values of EBC, $V_f$, and $V_c$ up to approximately 43° S; (2) a blurred maximum or a plateau in the latitude zone of 44–56° S; and (3) and a rapid drop up to the Antarctic coast. While moving from Africa to Antarctica, the volume of aerosol particles, on average, decreased from 5.5 μm³/cm³ to 0.55 μm³/cm³, and the concentration of EBC decreased from 68.1 ng/m³ to 17.4 ng/m³. The decrease in the aerosol content at high latitudes (φ > 55° S) was affected additionally by ice and icebergs, inhibiting the sea aerosol generation. A consequence of this was the more rapid decrease observed in the content of large particles.

The latitude variations in AOD and EBC calculated using MERRA-2 reanalysis data were also characterized by the average values decreasing from Africa up to Antarctica: AOD decreased by a factor of 2.3, and EBC decreased by a factor of 4. However, important deviations from the actual measurements existed. The curve of the average latitude behavior of the model AOD values was above that from the ship-based measurements and showed a maximum at the latitude of 48° S. On the contrary, the average model-based EBC (φ) values were 20–40% below the ship-based measurements. Still, another difference between the model- and ship-based data was manifested in the interannual variations in the summer (December–February) values of AOD and EBC over the Southern Ocean (φ > 60° S). The MERRA-2 reanalysis data showed a positive trend, inconsistent with the data from the actual AOD measurements over the Southern Ocean and at the Mirny observatory.

An analysis of the spatial variations in the aerosol characteristics near the Antarctic islands revealed an interesting effect: the volumes of particles in both fractions increased by a factor of 2.5, while the concentrations of EBC statistically significantly decreased by 29% at a distance of 300 km from the islands. The growth of the particle volumes may have been due to the more intense generation of marine aerosol in the open ocean. The

reason why the concentrations of EBC behaved in an opposite way is yet unclear. During the measurements near the islands, there were no vegetation burning and anthropogenic impacts (from heating systems) that could lead to the increased concentrations of EBC. It was hypothesized that the source of light-absorbing aerosol on the islands was humic substances, which were formed by the decomposition of the products of life activities in large colonies of penguins, pinnipeds, and birds, as well as algae.

A common feature of the spatial distribution of the ion compositions of aerosol was that the ion concentrations decreased from Africa up to Antarctica. Against the background of this common regularity, a minor maximum (growth of the concentrations) at the latitude of $40°$ S was manifested in most ions (except in $NH_4^+$ and $NO_3^-$). The largest latitude variability range was observed in the ions $Cl^-$, $Na^+$, and $Mg^{2+}$: their average concentrations decreased by a factor of 10–15. The concentrations of the ions $SO_4^{2-}$, $Ca^{2+}$, and $K^+$ changed by a factor of 4–6. The behaviors of the concentrations of $NH_4^+$ and $NO_3^-$ diverged from a common regularity: their latitude variations were smaller than the standard deviations in separate latitude zones. The ion concentrations lined up in the following order: $Cl^-$, $Na^+$, $SO_4^{2-}$, $Mg^{2+}$, $Ca^{2+}$, $K^+$, $NH_4^+$, and $NO_3^-$, remaining unchanged at all latitudes.

An analysis of the fractional factors showed that the main contribution to the ion composition of aerosol in most of the study region was due to marine sources: $FM_{sea} \approx 0.88$. This is also evident from the enrichment coefficients: the $K_i$ value for the ions $Cl^-$ and $Mg^{2+}$ was close to unity (the ion content was the same as that in seawater) at all latitudes, and the enrichment coefficients for ions $SO_4^{2-}$, $Ca^{2+}$, and $K^+$ in the latitude zone of $35$–$57°$ S also showed low (0.8–2.2) values. With decreasing distance to Antarctica, the fractional factor $FM_{cont}$ increased to 0.34, while the enrichment coefficients for ions $SO_4^{2-}$, $Ca^{2+}$, and $K^+$ increased to 4–6. This growth of the $FM_{cont}$ and enrichment coefficients indicated the increasing effect of the sources on the Antarctic coast, coupled with the weakening contribution from marine sources (increased area of the ice-covered surface of the ocean and lower water salinity).

**Supplementary Materials:** The following supporting information can be downloaded at: https://www.mdpi.com/article/10.3390/atmos13030427/s1, Figure S1: Histograms of data distribution by months of measurements; Table S1: Number of days of the measurements of different characteristics and collected aerosol samples (n) in the latitude zone of $34$–$70°$ S of the Indo-Atlantic sector of the ocean ($45°$ W–$110°$ E).

**Author Contributions:** Conceptualization and writing—original draft, S.M.S.; Organization of expeditionary measurements, V.F.R.; Expeditionary measurements, Y.S.T. and O.R.S.; Chemical analysis of samples, L.P.G. and O.I.K.; Processing, analysis, and interpretation of data, D.M.K., V.V.P. and L.P.G. All authors have read and agreed to the published version of the manuscript.

**Funding:** This work was supported by the Russian Science Foundation (Grant No. 21-77-20025).

**Institutional Review Board Statement:** Not applicable.

**Informed Consent Statement:** Not applicable.

**Data Availability Statement:** Not applicable.

**Acknowledgments:** The AOD data were obtained using techniques and equipment of the Center for Collective Use "Atmosphere" with support from the Ministry of Education and Science of Russia (Agreement No. 075-15-2021-661). The authors thank their colleagues who participated in the measurements and preparation of instrumentation, i.e., A. V. Gubin, M. I. Grachev, K. E. Lubo-Lesnichenko, Vas. V. Polkin, A. N. Prakhov, D. E. Savkin, A. P. Rostov, V. P. Shmargunov, S. A. Terpugova, A. B. Tikhomirov, S. A. Turchinovich, N. I. Vlasov and P. N. Zenkova. We also thank the organizers of the site https://giovanni.gsfc.nasa.gov/giovanni, accessed on 15 January 2022. for the opportunity to use important information.

**Conflicts of Interest:** The authors declare no conflict of interest.

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
