# Peer review of "Spatial Distribution of Aerosol Characteristics over the South Atlantic and Southern Ocean Using Multiyear (2004–2021) Measurements from Russian Antarctic Expeditions"

_atmosphere, doi:10.3390/atmos13030427_

Round 1
Reviewer 1 Report
The manuscript „Spatial Distribution of Aerosol Characteristics Over the South Atlantic and Southern Ocean, Using Multiyear (2004-2021) Measurements in Russian Antarctic Expeditions” summarises a long period of aerosol observations in the Southern Ocean, leading to a new understanding of temporal and spatial patterns in aerosol optical depth, elemental black carbon and giving some insights into the ionic composition of the aerosol in the area. The long data series and a broad scope of the conducted analysis recommend the article for publication in Atmosphere. It i salso rather well-written (minor editorial comments are listed below). However, I believe some minor changes are still necessary prior to publication, and one important change, which is adding the actual Conclusion to the manuscript.
(Please note that since lines are not numbered, it has been difficult to refer to specific parts of the text, therefore some phrases will be quoted without giving their exact location in text.)
General comments:
- Please make sure the whole manuscript is written in English – there remains the „и” letter intead of „and” in two places, as well as there is a whole paragraph first written in Russian, and then repeated in English.
- please make sure to spell out all abbreviations at their first mention in the text, as well as in figure and table captions wherever they occur again; I think „EBC” should be spelled out in full (including that it means „elemental black carbon” or the equivalent of elemental black carbon); there is also a typo in the abbreviation AOD in one place (OAD);
- whenever you refer to black carbon, I think it should be named „light-absorbing” instead of just „absorbing” substance (clearly, you do not mean its chemical properties, but optical);
Specific comments:
- Introduction
„The spatial distribution of, properly, marine aerosol over the ocean is relatively uniform.” - I would skip the word „properly”, the meaning is confused by it
„The geographic differences may be due to the specific features of the hydrometeorological conditions of generation of ma-rine aerosol; but, the main effect is due to the outflows of continental (dust, anthropo-genic, and smoke) aerosol to the marine atmosphere. That is, the spatial nonuniformities of aerosol over ocean are determined primarily by the strength of the continental sources in adjoining regions and the predominant air mass circulations. The strongest effect is exerted by the trade-wind and monsoon outflows of aerosols of different types in the At-lantic, Pacific, and Indian Ocean basins [e.g. 4-7].” - do refs 4-7 pertain to the whole excerpt? Please add some to the sentence before if not.
- Characterisation of expedition measurements
The ion chromatography analysis method should be reported with full analytical QA/QC data (precision and accuracy errors, analytical range, LOD and LOQ). Quoting another publication on the method is not sufficient.
- Results
This section would be more accurately named „Results and discussion”
„which overlap the source of marine aerosol” – the wording „overlap” would be clearer replaced by e.g. „limits” or „blocks” (the word occurs again in text elsewhere)
„In this case, we kept ourselves from quantitative comparison, because the publications by other authors present average aerosol characteristics for just separate expeditions in the Southern Ocean, which can differ from our multiyear data.” – it may be interesting, though, to know how far such differences may reach
„latitude decrease up to the Antarctic coasts” – please replace by „latitude decrease towards the Antarctic coasts” (repeats elsewhere)
„The latitude dependences of the black carbon concentrations show an opposite relationship between the model and experimental data (Figure 5а) …” - Could surface albedo change play a role in the EBC MERRA 2 data, too?
„The average difference between the data Δ = [ a0.5 (MERRA-2) – [ a0.5 (RV)] had been 0.015.” – why not report mean square error as well, for clarity?
„but the average difference is comparable to the error of measuring and modeling these characteristics” - Where is this error quantified (reference)?
„in the Northern Ocean” - that sounds like the Arctic Ocean; please refer to latitudes instead
„normalized* values”, and further on with averages - please put the asterisk in brackets and say "marked with an asterisk"; as it is, it seems like a mark for a footnote
„It is logic to speculate” – logical
„the period of the every aerosol sampling” - did you mean "in every period of aerosol sampling"?
„In order not to overburden the plots of the latitude behaviors of Ci (φ), the error bars, indi-cating the standard deviations (SD), are shown for just one variant of the calculation (b; Δφ = 3°) in Figure 11а and only for the upper and lower dependences Ci (φ) in Figure 12.” - OK, but please comment here on whether these missing error bars would be similar in values to the ones shown
„At the same time, there are doubts about the significance of the features manifested, because the latitude variability ranges of ions NH4+ and NO3- are narrower than the standard deviations in separate latitude zones.” - Still, does the sharp change in nitrate concentrations after 55 deg. S have any likely reason?
„in ions of marine origin” - Typically or inferred here? Because some of the other ions may also come from sea-salt in significant proportion.
„(primarily, the Na+ and Cl-)” - Maybe clearer "primarily represented by the Na+ and Cl- ions"?
„The smaller FMcont values near Afri-ca, as compared to Antarctica, could be due to the methodic factor of sampling.” - Another factor is that the FMcont is a relative contribution, hence if there is less marine aerosol near Antartica than near Africa, it will also produce higher FMcont values from the same absolute continental aerosol loads. How does this influence the overall result?
Finally, there are some interpretation questions which could be addressed additionally:
- Have you considered dark rock dust as an alternative source?
- Have you considered algal blooms producing DMS as a potential source of sulphate aerosol?
- Have you checked if there is an island effect for the biogenic ions (ammonium and nitrate) to cross-check the biological brown carbon hypothesis? It is not a certain tell yet perhaps it will contribute at least to tracing the extent of influence of the penguin rookeries.
It would be also good to address briefly, either at the end of Discussion, or in Conclusions, what are the limitations of this study.
- Conclusions
This section is more summary than conclusions. Maybe rename the section to "Summary and conclusions". Still, the actual conclusions need to be added, i.e. further research ideas, remaining questions, significance of the findings as a whole (in light with other research).
Author Response
We thank the Reviewer for the analysis of our manuscript and useful comments. We answered all questions and comments. Answers are presented in the PDF file.

Reviewer 2 Report
In this study, the authors aggregate aerosol measurements from successive cruises across the Southern Ocean to build a climatology of properties in a region that is difficult to characterize remotely. The measured characteristics decrease with decreasing aerosol concentrations closer to Antarctica.
The paper is satisfactorily written, although it could do with editing for English language. Figures illustrate the discussion well. I suggest removing a few speculative aspects of the discussion that are not well supported by the evidence, as listed as main comments below. Improvements to the discussion are also needed in places. For these reasons, I recommend minor revisions because there should be no need for further data analysis.
Main comments:
- The second paragraph of page 9, beginning with “This transformation of…”, is speculative. Can the authors provide stronger supporting evidence? There is a sizeable fine mode in marine aerosols too, and I cannot see why that would be transported much further than the coarse mode. Removal must be very efficient across all sizes.
- Most of section 3.2 is unconvincing. Causes for disagreement between MERRA-2 and cruise measurements can be many, and the observations are not an actual climatological average, so sampling is very different, as acknowledged by the authors. I would strongly shorten the section – the first paragraph on page 11 and Figure 8 are the only one worth keeping, because they do the comparison properly.
- Section 3.4 is also unconvincing. The data shown on Figure 10 is extremely noisy. The relationship depending on distance claimed by the author is unlikely to be statistically robust. Even if it was real, can coastal zone effects (last paragraph of page 14) be strong enough to explain it? What kind of absorbing compound would support the theory given in the last paragraph of section 4.2 on page 15? The authors would need to provide stronger supporting evidence, but I do not think there is a relationship in the first place, so the whole section could be removed.
Other comments:
- Abstract: “where the aerosol content corresponds to the global background level” and similar statements in first paragraph of page 2. What is meant by “background” here? I would expect Antarctica to be much cleaner than any global background level, if there is such a thing. The concept of background level is really only useful globally.
- Bottom of page 1: “ocean is relatively uniform.” This is contradicted a few sentences later, so there is no need to say that here.
- Page 2: “A change of this level serves as an indicator of global changes in aerosol content in the Earth’s atmosphere such as was the case after strong volcanic eruptions of El Chichón and Pinatubo.” This refers to increases in stratospheric aerosols, which seem out of scope of this study. I suggest deleting that statement.
- Second to last paragraph of page 2: typo OAD
- First paragraph on page 5: Isn't there a risk of filtering out instances of continental aerosol transport? And “false measurements” is not really accurate. “Contaminated measurements” perhaps?
- Page 6, first paragraph of section 3: “the generation of marine aerosol, which depends on the wind velocity”. There are other dependencies: temperature, algae, etc.
- Figure 5a: Why are the error bars one sided?
- Second paragraph of page 8: “thus favoring a certain aerosol accumulation”. But it is not just about circulation, one needs to consider removal too. I think stronger evidence is needed to argue for an “accumulation” at those latitudes.
- Section paragraph of page 10: “showed their qualitative agreement.” There is no agreement for AOD, even qualitative, as stated in the next sentences of the same paragraph.
- First paragraph of page 11: “the error of measuring and modeling these characteristics.” How did you quantify those errors?
- Paragraph on pages 11/12 is in Russian…
Author Response
We thank the reviewer for his consideration of this manuscript and drawbacks revealed. Most of the drawbacks were corrected, and all questions answered. However, we don’t quite understand why it is so categorically proposed to exclude a few sections of our research. The reasons for this could be: (а) the plagiarism or the absence of novelty in our results; and (b) the inconsistency with the modern physiochemical understanding of aerosol. The novelty does exist because the materials of the manuscript are still the only statistical generalization of the aerosol characteristics over the Southern Ocean (more specifically, in the region of 34° – 70°S; 45°W – 110°E), based on the multiyear data of the yearly measurements (20 expeditions in 2004-2021). No facts, confirming that our results are contradictory, are either provided. We present the answers to the questions and responses to comments. Answers are presented in the PDF file

Reviewer 3 Report
This manuscript documents the spatial distribution of aerosol characteristics over the South Atlantic and the Southern Ocean, using multiyear (2004-2021) measurements in Russian Antarctic Expeditions. The topic of the paper is of general interest for both the scientific world and for development.
General comments:
- The paper is well written although the style of writing makes the message unclear.
- The lines were not numbered, making it difficult to review and refer to where problems are.
- Please make sure all abbreviations are fully explained on their first use in the manuscript
Minor reviews:
- What is the full meaning of SPM on page 2, paragraph 3
- On page 2, paragraph 4, you used the abbreviation OAD. Do you mean AOD??
- What is the meaning of RV on page 3 under section 2?
- Parts of pages 11and 12 are written in Russia. What happened?
Author Response
We thank the Reviewer for the positive assessment of our manuscript and helpful comments. We have answered all questions and comments.
Answers are presented in the PDF file

Reviewer 4 Report
The authors reported the spatial distribution of physicochemical aerosol characteristics over the South Atlantic and Southern Ocean, with multiyear (2004-2021) measurements in Russian Antarctic Expeditions, which is quite impressive. This type of the study is important for the scientific society. However, the manuscript was not well prepared. Additionally, the language requires substantial improvement (both style and grammar) throughout the manuscript. Many sentences are not clearly written. There are quite a few major issues with the study that prevent me from recommending it for publication in the present format. It is possible that these issues could be addressed with a major revision. My specific concerns are addressed below.

Author Response
We thank the Reviewer for the analysis of our manuscript and useful comments. We answered all questions and comments.
Answers are presented in the PDF file

Round 2
Reviewer 4 Report
The manuscript has been improved. I recommend publication of this manuscript.This manuscript is a resubmission of an earlier submission. The following is a list of the peer review reports and author responses from that submission.